# Low-Level Laser Irradiation Stimulates RANKL-Induced Osteoclastogenesis via the MAPK Pathway in RAW 264.7 Cells

Jae-Min Song [1,2], Bong-Soo Park [2,3], Sang-Hun Shin [1,*,†] and In-Ryoung Kim [2,3,*,†]



1   Department of Oral and Maxillofacial Surgery, School of Dentistry, Pusan National University, Yangsan 50612, Korea; songjm@pusan.ac.kr

2   Dental and Life Science Institute, School of Dentistry, Pusan National University, Yangsan 50612, Korea; parkbs@pusan.ac.kr

3   Department of Oral Anatomy, School of Dentistry, Pusan National University, Busandaehak-ro, 49, Mulguem-eup, Yangsan 50612, Korea

*   Correspondence: ssh8080@pusan.ac.kr (S.-H.S.); biowool@pusan.ac.kr (I.-R.K.); Tel.: +82-51-510-8552 (I.-R.K.)

†   Both the authors contributed equally to this study.

**Abstract:** Low-level laser therapy (LLLT) is recognized as an effective medical tool for the treatment of various conditions requiring tissue repair, pain relief, inflammation treatment, and restoration of tissue dysfunction, and its development and research are growing rapidly. However, studies that analyze molecular biology by applying LLLT to osteoclasts are still insufficient to understand the mechanism. In order for LLLT to be suggested as an appropriate treatment method for the treatment of various bone diseases, it is necessary to elucidate the effect and mechanism of LLLT on osteoclast differentiation. In this study, we investigated the effect of LLLT on osteoclast differentiation using murine macrophage (RAW 264.7) cells by means of a Ga-As-Al laser ($\lambda$ = 810, 80 mW). Our results indicate that LLLT did not induce cytotoxicity in RAW 264.7 cells. When LLLT was applied for 15 s to osteoclasts exposed to RANKL, the expression of NF-$\kappa$B, ERK, p38, and c-Fos, which are associated with expression of NFATc1, was increased. The RT-PCR results also demonstrated significantly increased expression of osteoclast-specific genes, including NFATc1, TRAP, the calcitonin receptor, and cathepsin K, compared with the control. Taken together, we concluded that low-level laser irradiation induces osteoclastogenesis by enhancing the expression of NF-$\kappa$B, MAPKs (ERK, p38), c-Fos, and NFATc1 in RAW 264.7 cells. These findings indicate that low-level laser irradiation could be considered a potential treatment option in various metabolic bone diseases that require osteoclastic activity and bone formation.

**Keywords:** low-level laser therapy (LLLT); osteoclastogenesis; MAPK pathway

## 1. Introduction

Bone homeostasis is a complex process that is precisely maintained throughout life by a balance between the bone-forming activity of osteoblasts and the bone-resorbing activity of osteoclasts [1–3]. If the equilibrium of this activity is broken, metabolic bone disease develops, which results in changes in bone formation and bone resorption [4]. Most metabolic bone diseases (osteoporosis, osteomalacia, etc.) are caused by increased osteoclast activity; conversely, osteopetrosis is induced as a result of decreased osteoclastic activity [5]. Therefore, research into the role of osteoclasts in metabolic bone disease is very important in overcoming disease.

Differentiation of bone marrow progenitor cells into multinucleated osteoclasts is induced through several complex processes involving the fusion of mononuclear progenitor cells [6,7]. The receptor activator of the nuclear factor $\kappa$B (NF-$\kappa$B) ligand (RANKL) is a cytokine produced by osteoblasts, which binds to the RANK (NF-$\kappa$B receptor activator) receptor present in osteoclast progenitor cells. As a result, this leads to the differentiation of osteoclast progenitor cells and the activation of mature osteoclasts [8,9]. The binding of

RANKL and its receptor RANK on the osteoclast induces the recruitment of cytoplasmic tumor necrosis factor receptor-associated factor 6 (TRAF6) activation, and this leads to the activation of downstream molecules, such as NFATc1, NF-κB, mitogen-activated protein kinases (MAPKs), c-Fos, and c-Jun [10–13]. These transcription factors induce the complex responses of various genes, allowing the osteoclast progenitor cells to mature into multi-nuclear osteoclasts [14]. Mature osteoclasts degrade bone matrix proteins and inorganic components of bone by releasing various types of protein, including: Calcitonin receptor, tartrate resistant acid phosphatase (TRAP), cathepsin K, and matrix metalloproteinase-9 (MMP-9) [15,16].

Among medical applications using lasers, low-level laser therapy (LLLT) is recognized as an effective medical tool, and its development and research are growing rapidly [17,18]. Medical applications of LLLT include the treatment of a multitude of conditions that require tissue repair, pain relief, inflammation treatment, and the restoration of tissue dysfunction [19]. When LLLT irradiates a living body, the photon LLLT light source is absorbed by intracellular pigments or photoreceptors to produce the following effects: Increased capillary production, cartilage production, nerve cell regeneration, muscle regeneration with relief of muscle atrophy, reduction in skin inflammation and swelling, and the promotion of collagen and bone formation [20–23]. It was found that ATP synthesis and cytochrome c oxidase activity increased after LLLT irradiation in isolated mitochondria. This means that most of the energy-absorbing molecules for light are proteins, many of which are elements found in mitochondria, such as cytochromes [24]. Although dental applications of LLLT are not well documented compared to musculoskeletal applications, studies have recently been conducted in several major dental specialties, including endodontic, periodontal, and orthodontic treatment, and in maxillofacial surgery [25]. In particular, LLLT has been shown to be effective in managing chronic pain-related hard and soft tissue lesions in the maxillofacial region [26,27]. Despite many laboratory and clinical reports, it is still controversial due to uncertainty about the underlying molecular and cellular mechanisms by which photons in LLLTs affect cells, i.e., the physical signals of light transformed into biological signals [28–30].

Numerous studies have reported that LLLT stimulates osteoblasts or osteoblast-like cells and has a positive effect on bone regeneration [31–34]. However, few studies have analyzed the molecular biology involved when applying LLLT to osteoclasts, and conclusions presented in studies have been controversial [35]. Therefore, in this study, we investigated the effect of LLLT on osteoclast differentiation using RAW 264.7 cells, and confirmed, at the cellular level, how much energy is appropriate when applying LLLT.

## 2. Materials and Methods

### 2.1. Cell Culture

The RAW 264.7 (murine macrophage) cell line was purchased from the American Type Culture Collection (ATCC; Rockville, MD, USA). RAW 264.7 cells were cultured in Dulbecco's modified Eagle's medium (DMEM, Hyclone, Logan, UT, USA) supplemented with 10% heat-inactivated fetal bovine serum (FBS, Gibco BRL, Grand Island, NY, USA) without antibiotics at 37 °C in a 5% $CO_2$ incubator. For the osteoclast differentiation, cells were divided into five groups and cultured under the RANKL (R&D Systems, Minneapolis, MN, USA) (10 ng/mL) and LLLT treatment (Table 1).

**Table 1.** LLLT irradiation parameters and RANKL treatment.

| Groups | LLLT Exposure Time | Total Energy Density | RANKL Treatment (10 ng/mL) |
|---|---|---|---|
| Group 1 | 0 s | - | - |
| Group 2 | 0 s | - | + |
| Group 3 | 5 s | 0.4 J/cm$^2$ | + |
| Group 4 | 15 s | 1.2 J/cm$^2$ | + |
| Group 5 | 30 s | 2.4 J/cm$^2$ | + |

### 2.2. Low-Level Laser Irradiation In Vitro

LLLT irradiation was performed with a gallium–arsenide–aluminum (Ga-As-Al) laser device (NDLux; Seoul, Korea). The tip of the laser device was fixed by using a clamp at a distance of 5 cm from the bottom of the plate. LLLT irradiation was at a wavelength of 810 nm and a maximum power of 80 mW continuous mode was delivered to the cells; the method was described previously [36]. LLLT parameters are listed in Table 1.

### 2.3. Cell Proliferation Assay

Cells ($1 \times 10^4$) were placed in a 96-well microplate and irradiated for 3, 6, or 24 h using LLLT. After the cells were treated with the Cell Proliferation Kit I (MTT, 3-[4,5-dimethylthiazol-2-yl]-2,5-diphenyltetrazolium bromide) (Sigma, St. Louis, MO, USA), they were incubated the 96-well plate for 4 h in a humidified atmosphere. The experimental method was carried out according to the manufacturer's instructions. The media were aspirated, and purple formazan crystals were dissolved in solubilization solution. The proliferation rates of completely dissolved purple formazan crystals were measured by a microplate reader (Tecan, Männedorf, Switzerland) with an absorbance wavelength of 570 nm.

### 2.4. TRAP Staining

Cells ($1 \times 10^4$ cells/well) were plated in a 12-well tissue culture plate in the presence of RANKL (10 ng/mL) and incubated for 5 days, and the medium was changed every day. Cells were then fixed in 4% formaldehyde solution (Sigma, St. Louis, MO, USA) for 10 min, washed with phosphate-buffered saline (PBS) solution twice, and permeabilized with 0.1% Triton X-100 for 10 min. After that, the cells were treated with the tartrate-resistant acid phosphatase (TRAP) activity kit (Sigma, St. Louis, MO, USA), according to the manufacturer's instructions, and then the cells were washed 3 times with distilled water. Cells containing three or more nuclei were classified as osteoclasts and counted.

### 2.5. Resorption Pit Assay

Cells ($1 \times 10^4$ cells/well) were seeded on a BioCoat Osteologic Bone Cell Culture System plate (BD Biosciences, San Jose, CA, USA), then LLLT was applied. Then, cells were cultured in a medium containing RANKL for 5 days, and the medium was changed every day. After incubation, the cells were lysed using 1N NaOH and the reabsorbed pits were photographed using an optical microscope (Olympus CX41, Shinjuku, Tokyo, Japan), and the area of the pits was measured and calculated.

### 2.6. Western Blot Analysis

After applying LLLT and RANKL to the cells for each group, the cells were harvested, RIPA (Invitrogen, Carlsbad, CA, USA) buffer was used to separate proteins, and the lysate (total protein) concentrations were determined by the Bradford protein assay (Bio-Rad, Richmond, CA, USA). Bovine serum albumin (BSA, Sigma, St. Louis, MO, USA) was used as a protein standard. The above method followed the manufacturer's instructions. Twenty micrograms of total protein from each group was separated using SDS-PAGE gel electrophoresis and transferred to a PVDF membrane (Amersham GE Healthcare, Little Chalfont, UK). After blocking in 5% nonfat dry milk for 1 h, the following antibodies were applied for 2 h at room temperature: Anti-ERK, anti-phosphorylated ERK, anti-JNK, anti-phosphorylated JNK, anti-p38, anti-phosphorylated p38 anti-NFATc1, anti-c-Fos, anti-GAPDH, peroxidase-conjugated goat anti-rabbit antibody. All antibodies were purchased from Cell Signaling Technology (Danvers, MA, USA). SuperSignal™ West Femto Maximum Sensitivity Substrate (Thermo Fisher Scientific, Waltham, MA, USA) was used as an immunodetection reagent and protein expression was quantified using an Alpha Imager HP chemiluminescence detection system (Alpha Innotech, Santa Clara, CA, USA)

### 2.7. Immunofluorescent Staining

Cells ($1 \times 10^4$ cells) were plated on a cover slip, and LLLT (15 s) and RANKL were applied, followed by incubation for 24 h. Cells were fixed in 4% formaldehyde solution at room temperature for 10 min and then permeabilized with Triton X-100 for 10 min. Next, the actin filaments of the cells were stained in red using the fluorescent molecule rhodamine–phalloidin (Invitrogen, Carlsbad, CA, USA). The samples were then incubated with anti-NF-κB (p65) (Cell Signaling Technology, Danvers, MA, USA) overnight at 4 °C. The next day, cells were washed with PBS, incubated with Alexa 488 (green fluorescence) conjugated secondary antibody (Enzo Life Sciences, Farmingdale, NY, USA) for 1 h, and, finally, cells were mounted using ProLong® Gold antifade reagent (Invitrogen, Carlsbad, CA, USA). Fluorescence images were observed and analyzed using an LSM 700 confocal microscope (Carl Zeiss, Göettingen, Germany).

### 2.8. RNA Isolation and RT-qPCR

Total RNA was isolated from the RAW 264.7 cells treated with LLLT and RANKL using an RNeasy mini kit (Qiagen Inc., Valencia, CA, USA) and RNA (2 μg) was reverse-transcribed using a RevertAid First-Strand Synthesis System kit (Thermo Fisher Scientific, Pittsburgh, PA, USA) to synthesize cDNA. The above manufacturer's protocols and methods have been described previously [36]. The gene expression was quantified by RT-qPCR using a SYBR Green kit (Applied Biosystems, Warrington, UK) on the ABI 7500 Real-Time PCR Detection System (Applied Biosystems, Foster City, CA, USA). The primers used in this experiment are as follows; NFATc1, forward: 5′-CCAGCTTTCCAGTCCCTTCC-3′ and reverse: 5′-AGGTGACACTAGGGGACACA-3′; TRAF6, forward: 5′-ATATGACAGCCACCTCCCCT-3′ and reverse: 5′-GGCAAGCAGTTCTGGTTTGG-3′; TRAP, forward: 5′-CTACCTGTGTGG ACATGACCA-3′ and reverse: 5′-GCACATAGCCCACAC CGTTC-3′; cathepsin K, forward: 5′-TACCCATATGTGGGCCAGGA-3′ and reverse: 5′- TTCAGGGCTTTCTCGTTCCC-3′; calcitonin receptor, forward: 5′-TAGTTAGTGCTCCTCGGGCT-3′ and reverse: 5′-AGTACTCT CCTCGCCTTCGT-3′; GAPDH, forward: 5′-CCGCATCTTCTTGTGCAGT-3′ and reverse: 5′-TCAATGAAGGGGTCGTTGAT-3′. Gene expression was calculated using the comparative Ct method and quantified with the housekeeping gene GAPDH to calculate Ct values for all samples.

### 2.9. Statistical Analysis

All experiments were repeated in at least three independent experiments. Results are expressed as mean ± S.D. Statistical analysis was performed using one-way ANOVA (GraphPad Prism 5, GraphPad Software Inc., San Diego, CA, USA) for cell viability and RT-PCR. $p$ values of less than 0.05 were considered statistically significant.

## 3. Results

### 3.1. LLLT Enhances RAW 264.7 Cell Proliferation, Osteoclast Differentiation, and Bone Resorption

The effect of LLLT on the survival or proliferation of RAW 264.7 cells was investigated through MTT assays. The RAW 264.7 cells were exposed to LLLT ($\lambda$ = 810 nm, 80 mW, 80 mA) for 5, 15, or 30 s, then incubated for 3, 6, or 24 h. In the groups cultured for 3 h and 6 h, there was no significant difference in any cells exposed to LLLT, but in the LLLT 30 s irradiation and 24 h incubation group, the cell viability was significantly increased ($p < 0.05$). Therefore, we confirmed that exposure to LLLT in RAW 264.7 cells does not negatively affect cell viability, but rather helps cell proliferation (Figure 1).

Next, we investigated whether LLLT enhances RANKL-induced osteoclast formation in RAW 264.7 cells. To confirm this, RANKL (10 ng/mL) was used to induce differentiation of the cells, and at the same time, LLLT was used for 5, 15, or 30 s, followed by incubation for 5 days. TRAP-positive multinuclear osteoclasts and resorption pit areas were identified using an optical microscope. It was confirmed that the number of TRAP-positive cells significantly increased (Figure 2A,B) in the LLLT 15 s irradiation group ($p < 0.05$). However, we observed a significant decrease in TRAP-positive cells in the LLLT 30 s irradiation group.

These results were also consistent with the pit formation (Figure 2C,D). Therefore, through these results, we confirmed that LLLT irradiation stimulates osteoclast differentiation after 5 to 15 s of treatment.

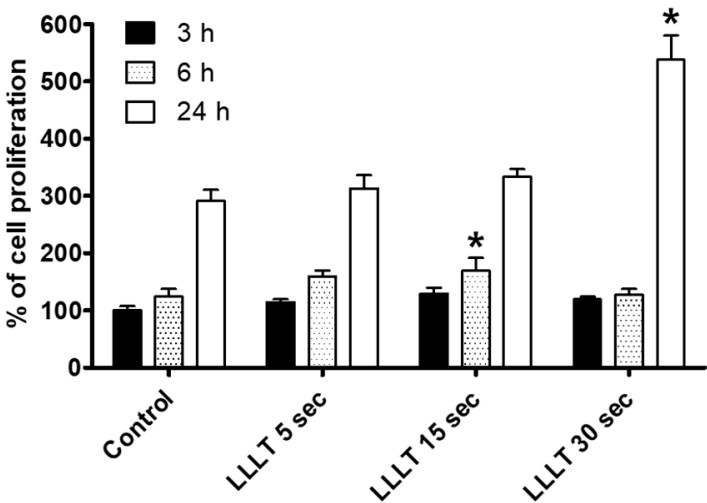

**Figure 1.** Cell proliferation affected by LLLT irradiation in RAW 264.7 cells. Cells were irradiated LLLT for 5, 15 and 30 s, then incubated for 3, 6 and 24 h. Cell proliferation measurement was conducted by MTT assay. Each value represents the mean ± S.D. * $p < 0.05$ compared with control.

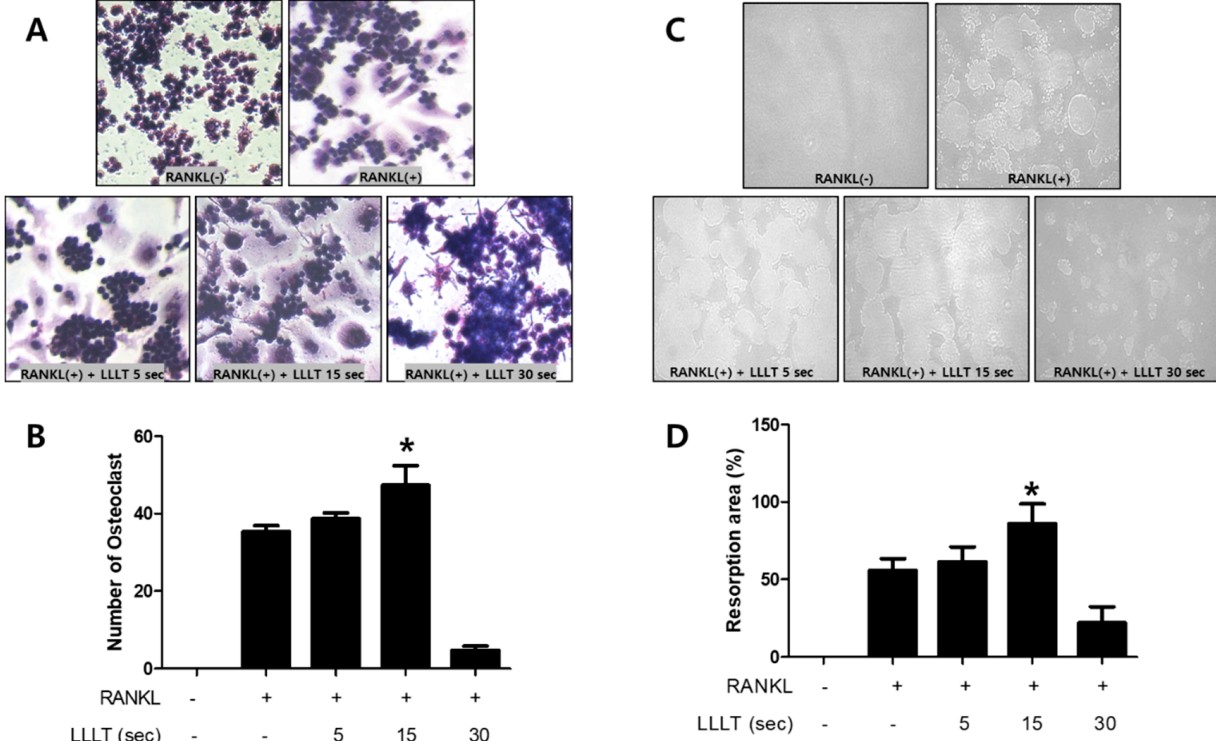

**Figure 2.** Effects of LLLT on RANKL-induced osteoclastogenesis and pit formation. (**A**) RAW 264.7 cells were cultured with RANKL (10 ng/mL) for 5 days to induce osteoclast differentiation with treatment with LLLT (5–30 s) to assess its osteoclastogenic effect. (**B**) The number of TRAP-positive multinucleated cells was counted. * $p < 0.05$ compared with RANKL alone. (**C**) Resorption pit formation were observed on Corning® Osteo Assay Surface, and (**D**) the pit area was calculated as the absorbed area/total area. * $p < 0.05$ compared with RANKL alone.

### 3.2. LLLT Shows a Difference in the Expression of Osteoclast-Forming Genes According to Irradiation Time

Real-time PCR was used to confirm changes in genes (NFATc1, TRAF6, cathepsin K, calcitonin receptor, and TRAP) related to osteoclast differentiation and according to the investigation of LLLT. As shown in Figure 3, the presence of RANKL and LLLT irradiation generally increased the gene expression patterns of NFATc1 and TRAP in RAW 264.7 cells. In particular, when LLLT irradiation was used for 15 s, the expression of all osteoclast-related genes significantly increased compared to the control group ($p < 0.05$). When comparing the expression of each gene in the RANKL only treatment group and the RANKL + LLLT irradiation group, the expression of NFAT-c1, TRAF6, TRAP, and calcitonin receptors was significantly increased when LLLT irradiation lasted for 15 s. These results suggest that even lasers of the same wavelength have different stimulating effects on osteoclast formation depending on the irradiation time.

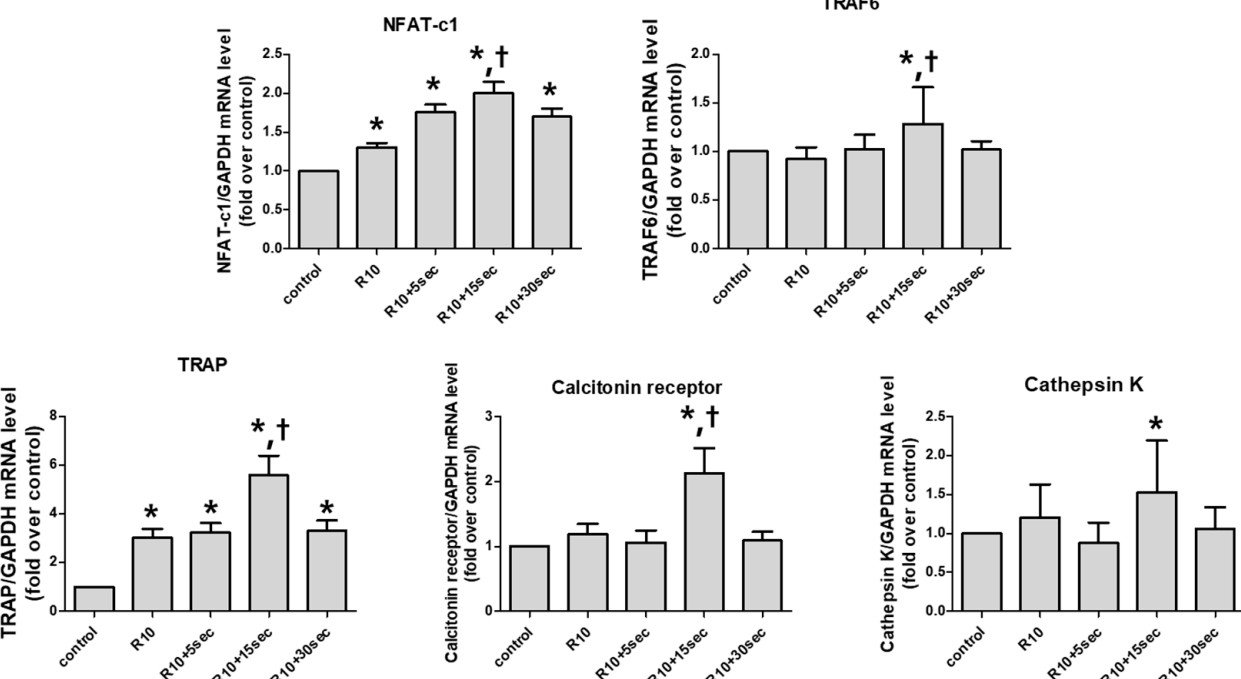

**Figure 3.** LLLT causes a difference in the expression of osteoclast-forming genes. RAW 264.7 cells were irradiated with LLLT (5~30 s) and incubated for 24 h in the presence of RANKL (10 ng/mL), then total RNA was extracted, and gene expression was analyzed by real-time RT-PCR. Each value represents the mean ± S.D. * $p <$ compared to control group; compared to RANKL alone, † $p < 0.05$.

### 3.3. LLLT Promotes the Transfer of NF-κB from the Cytoplasm to the Nucleus

To elucidate the molecular mechanisms underlying osteoclast differentiation, the activation of RANKL-induced nuclear translocation of p65, a subunit of NF-κB, was evaluated via confocal microscopy. RAW 264.7 cells were stained with DAPI (blue, nuclei) and Alexa 488 (green, p65). In the cells of the control group, p65 was located in the cytoplasm, and, in the cells treated with RANKL, p65 was confirmed to transfer from the cytoplasm to the nucleus. In addition, it was found that in the group treated with RANKL and LLLT, lasting for 15 s, the expression of p65 increased more than in the group treated with RANKL alone, and it was clearly transferred from the cytoplasm to the nucleus (Figure 4). Therefore, this result suggests that LLLT irradiation increases the expression of NF-κB in RAW 264.7 cells, promoting the transfer from the cytoplasm to the nucleus, thereby inducing the transcription of various genes.

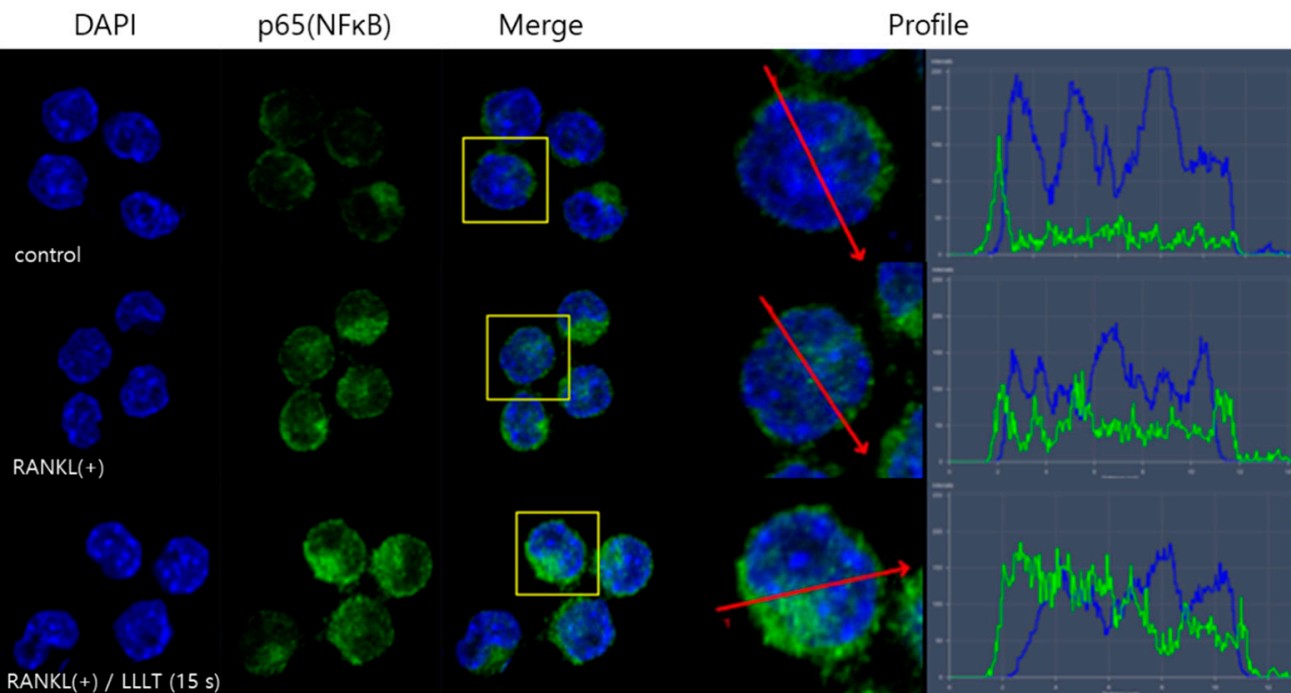

**Figure 4.** Effect of LLLT on NF-κB activation in RANKL treated RAW 264.7 cells. Using confocal microscopy, the positions of the nuclei stained with DAPI (blue) and the p65 protein stained with green fluorescence were confirmed. The location of p65 was expressed as a peak (right panel) through profiling.

### 3.4. LLLT Stimulates MAPK Pathways in Osteoclast Differentiation

Besides the NF-κB signaling pathway, activation of the MAPK pathway plays a pivotal role in osteoclastogenesis [37]. The next experiment was conducted to confirm the change in the expression of MAPKs due to irradiation by LLLT. RAW 264.7 cells were irradiated with LLLT for 15 s, and MAPK activity was confirmed over time from 15 to 120 min. The phosphorylation of ERK was not expressed, but the phosphorylated forms of JNK and p38 increased after 30 min (Figure 5A). In Figure 5B, the expression pattern of MAPK was confirmed 24 h after treatment with RANKL and LLLT. In the RANKL plus LLLT group, the phosphorylated forms of ERK, JNK, and p38 were significantly increased compared to RANKL or LLLT alone. It is known that c-Fos and NFATc1 are induced by p38 MAPK activation during osteoclastogenesis [1]. Phosphorylated c-Fos expression was maximal at 30 min post-LLLT treatment, while NFATc1 expression was increased at 120 min (Figure 6A). When LLLT was applied for 5 and 15 s on RAW 264.7 cells in a RANKL-containing medium and then incubated for 24 h, c-Fos and NFATc1 expression was increased compared to the control (Figure 6B). This result indicates that LLLT irradiation affects the p38 MAPK signaling pathway and activates c-Fos and NFATc1 transcription.

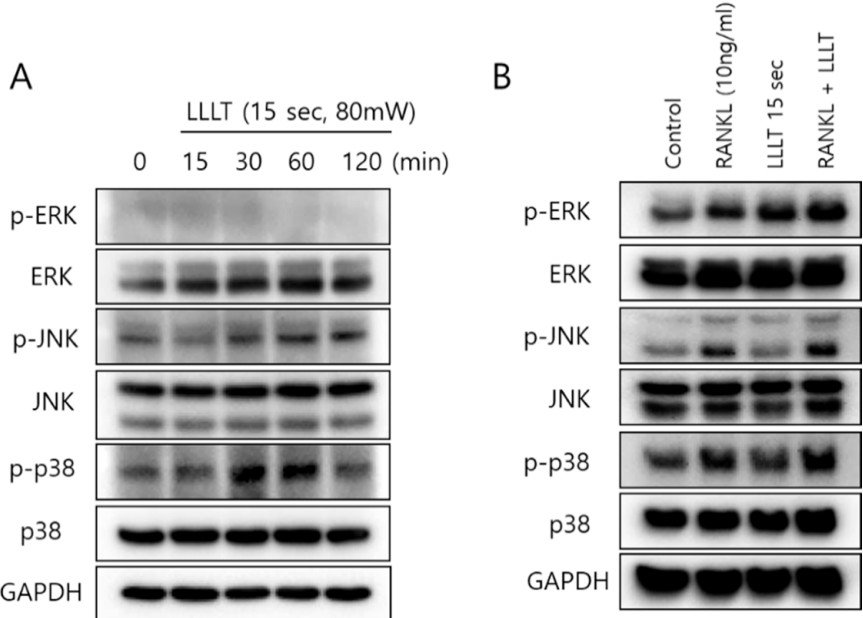

**Figure 5.** Effect of LLLT irradiation on the MAPK pathway in RAW 264.7 cells. (**A**) Only LLLT (15 Scheme 264.7 cells, and the change pattern of MAPK was confirmed for 15 to 120 min. (**B**) RAW 264.7 cells were pretreated with RANKL (10 ng/mL) and LLLT for 15 s and incubated for 24 h and the change in MAPK was confirmed.

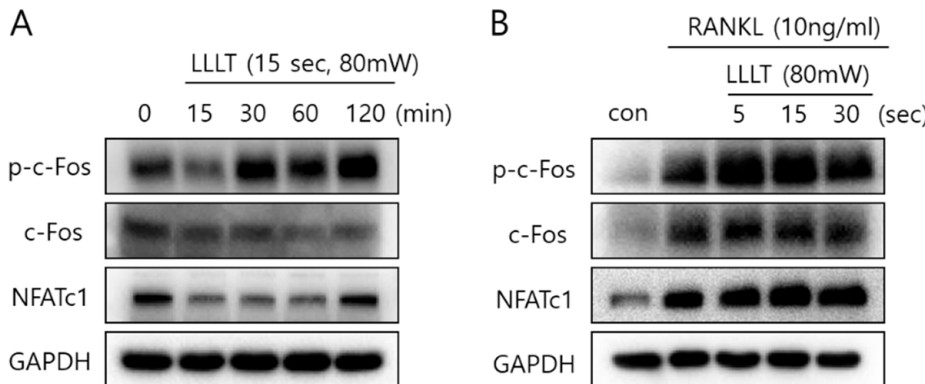

**Figure 6.** Activation effects of the c-Fos and NFATc1 pathways by LLLT irradiation. (**A**) Only LLLT (15 s) was applied to RAW 264.7 cells, and the change pattern of MAPK was confirmed for 15 to 120 min. (**B**) In the presence of RANKL, LLL was irradiated for 5 to 30 s, and after incubation for 24 h, the expression of c-Fos and NFATc1 was confirmed.

## 4. Discussion

Most studies of metabolic bone disease and bone healing investigate either activation of osteoblasts or suppression of osteoclasts [38,39]. However, enhancing or recovering osteoclastic activity and differentiation can also play an important role in bone remodeling [40]. Kawasaki reported that low-level laser irradiation enhanced tooth movement and the formation of osteoclasts on the pressure side in vivo [41]. Likewise, there have been a number of studies demonstrating the effectiveness of LLLT in bone healing, and there are some studies showing that it inhibits the differentiation of osteoclasts, but this is still controversial [19,35]. The purpose of this study was to investigate the effect of LLLT irradiation on osteoclast differentiation through a cytological approach to osteoclast differentiation using LLLT, which is widely used in bone healing.

The RAW 264.7 murine macrophages used in this study have been widely used in osteoclast-related studies due to various advantages, such as easy experimental approach,

rapid RANKL-mediated formation, large number of osteoclasts, and bone resorption (Figure 2) [42].

Mature osteoclasts differ from the progenitors of the hematopoietic/macrophage lineage in their cell-to-cell interactions with osteoblasts. RANKL binds to its receptor RANK on the precursor osteoclast and induces NFATc1 expression, which is a key regulator of osteoclast differentiation through signaling pathways. RANK recruits TRAF6 and activates NF-κB, while MAPKs and c-Fos result in NFATc1 activation (Figure 3) [43]. The functional importance of the NF-κB, JNK, and p38 MAPK pathways in osteoclastogenesis has been extensively studied [44,45]. NF-κB is a dimeric transcription factor. It is composed of p65 (RelA), c-Rel, RelB, NF-κB1 (p50), and NF-κB2 (p52) subunits. In vivo studies using p50- and p52-deficient mice suggest a critical role for NF-κB in osteoclastogenesis [46]. During osteoclast differentiation, the binding of RANKL and RANK activates inhibitors of the IκB kinase (IKK) complex in the NF-κB pathway, inducing IκB ubiquitination and proteolysis [47]. These events result in the translocation of the NF-κB dimer to the nucleus, enhancing transcription of the target gene. [12]. In this study, it was found that p65 was translocated from the cytoplasm to the nucleus in RANKL only treatment, while in the LLLT irradiation group, the expression of p65 increased and the translocation to the nucleus was more pronounced than in the RANKL only treatment (Figure 4). Therefore, we suggest that LLLT stimulates osteoclast formation by promoting NF-κB activation.

The effect of LLLT on ERK expression in osteoclasts is still unknown; however, Miyata et al. found that the MAPK/ERK pathway plays a role in the activation of dental pulp cells following low-level diode laser irradiation [48,49]. In this study, phosphorylation of ERK was maximal at 30 min after LLLT; maximum phosphorylation was also achieved after 15 s of LLLT exposure in the presence of RANKL as compared to either treatment alone. ERK expression was enhanced in osteoclastogenesis under LLLT. Laser irradiation appears to stimulate the expression of ERK, regardless of cell type.

JNK regulates a variety of cellular functions including cell growth, differentiation, survival, and cell death [50,51]. Ikeda et al. [52] found that blocking of the JNK pathway by JNK inhibitors inhibits osteoclast differentiation and that JNK is an essential signaling molecule for osteoclast differentiation. However, another study found that a substance called glutaredoxin2 isoform b (Glrx2b) promotes the osteoclast differentiation process, but the expression of JNK remains unchanged [53]. Thus far, the effects of LLLT on MAPK and osteoclast formation have shown that JNK expression is not induced in osteoblasts or skeletal muscle cells after Er:YAG laser and He-Ne laser treatment, respectively [54]. We found that in RAW 264.7 cells not treated with RANKL, JNK was not phosphorylated after LLLT treatment. However, in the group to which LLLT was applied after treatment with RANKL, phosphorylation of JNK was found. Based on other studies and our own results, we conclude that JNK is not affected by LLLT when it is in the mononuclear form, whereas in the case of cells whose osteoclast differentiation signal is initiated by RANKL, it is affected by LLLT. Like JNK, p38 MAPK is important during the early stages of osteoclast formation and has been shown to induce cathepsin K gene expression [55], while LLLT has been reported to enhance p38 MAPK signaling in osteoblast pulp cells and fibroblasts [48,54]. Research on osteoclast formation and p38 MAPK pathway by investigating LLLT is still lacking [56]. Collectively, our data suggest that LLLT stimulates osteoclast differentiation through the activation of ERK and the p38 pathways but not via JNK-containing pathways. Further studies will be needed to investigate the portions of the MAPK pathway that impact osteoclastogenesis after LLLT.

c-Fos, which forms a heterodimer with c-Jun, resulting in the formation of the activator protein-1 (AP-1) complex, induces NFATc1 activation during osteoclastogenesis [57]. Mice lacking c-Fos develop osteopetrosis as a primary pathology. It is rapidly and transiently induced within 15 min of stimulation in the fibroblast cell line by growth factors [58]. In this study, expression of c-Fos was maximal by 30 min post-LLLT treatment in the presence of RANKL. A variety of stimuli, including serum, growth factors, tumor promoters, cytokines, and UV radiation, are known to induce expression of c-Fos. Whether laser irradiation

affects the expression of c-Fos or not, LLLT stimulates the formation of osteoclast-like cells via RANK expression. In our data, c-Fos expression was enhanced when treated with LLLT for 15 s. These data suggest that LLLT stimulates osteoclast differentiation through the c-Fos pathway.

NFATc1 acts as a master modulator for osteoclast activation, fusion, and maturation [15]. The activation of NFATc1 is dependent on the transcription factors AP-1 (containing c-Fos) and NF-κB. NFATc1 translocates to the nucleus and induces various osteoclast-specific genes: TRAP, calcitonin receptor, and cathepsin K genes are all regulated by NFATc1 [59]. Taken together, these results indicate that the expression of NFATc1 and of osteoclast-specific genes is increased, suggesting that NFATc1 is activated by laser irradiation.

The mechanism by which LLLT promotes osteoclast formation is still unclear, but researchers have suggested several hypotheses. The effect of LLLT on cells is closely related to mitochondria, and it has been reported that it is mediated by the cytochrome c oxidase present therein [60]. It has also been reported that stimulation of cell proliferation by LLLT may be due to activation of signaling pathways such as the MAPK cascade or the PI3K/Akt pathway, which regulate the regulation of gene expression [61]. Aihara et al. [9] used a 50 mW, 810 nm wavelength laser with a 5 mm diameter fiber. They treated osteoblasts and osteoclasts at the 810 nm wavelength band, but reported that neither treatment had an effect on cell differentiation. Renno et al. [62] applied LLLT of various wavelengths (670, 780, or 830 nm) to normal primary osteoblast (MC3T3) and malignant osteosarcoma (MG63) cell lines, which are osteogenic cell lines. They concluded that each cell line responds differently to cell proliferation and ALP activity depending on the specific wavelength and dose combination. Fukuhara et al. [63] demonstrated that a single laser exposure at a wavelength of 905 nm of rat fetal calvaria cells increased osteoblast differentiation. In addition, Arisu et al. [64] investigated human osteoblast-like cells (Saos-2) using a 1064 nm wavelength Nd:YAG laser, and they revealed that the increase in pulse energy, pulse repetition rate, and power output negatively affects cell viability and proliferation. In the present study, the LLLT energy was 810 nm, 80 mW, 1.2 J/cm$^2$, which was lower than that of the LLLT used in the aforementioned reports. It is very difficult to find the optimal dose of LLLT in osteoclast formation due to the many factors involved, such as the different methodologies, different cell types, and different radiation times and distances. Bouvet-Gerbettaz et al. [65], when using LLLT with osteoblasts and osteoclasts, suggested setting up an LLLT investigation protocol using a wattmeter for LLLT treatment, as there may be significant differences between the actual emission power and the power applied to the cells. Taken together, we found that LLLT promotes osteoclast formation in RAW 264.7 cells through signaling pathways associated with NF-κB, MAPK (ERK, p38), c-Fos, and NFATc1. For patients with metabolic bone disease and various other diseases, the selection of an appropriate laser source and the standardization of radiation parameters are vital, and further studies are needed to optimize low-level laser treatment. Based on the results of this study, further exploration of this technique using different bone states, different LLLT parameters, and different follow-up periods will be of great value in helping to overcome bone diseases.

## 5. Conclusions

In this study, we investigated the effect of LLLT (Ga-As-Al laser, λ = 810, 80 mW) on osteoclast formation in vitro.

It was concluded that low-level laser irradiation promotes the expression of NF-κB, MAPK (ERK, p38), c-Fos, and NFATc1 in mouse macrophage RAW 264.7 cells more than under normal differentiation conditions to induce osteoclast formation. Therefore, this suggests that it can be considered as a potential treatment method for various metabolic bone diseases that require osteoclast activity and bone formation.

**Author Contributions:** Conceptualization, J.-M.S. and I.-R.K.; methodology, J.-M.S. and I.-R.K.; investigation, J.-M.S. and I.-R.K.; resources, B.-S.P. and S.-H.S.; data curation, J.-M.S. and I.-R.K.; writing—original draft preparation, J.-M.S. and I.-R.K.; writing—review and editing, S.-H.S. and I.-R.K.; visualization, B.-S.P. and S.-H.S.; supervision, B.-S.P. and S.-H.S.; funding acquisition, J.-M.S. All authors have read and agreed to the published version of the manuscript.

**Funding:** This study was supported by the Dental Research Institute (PNUDH DRI -2014-04), Pusan National University Dental Hospital.

**Institutional Review Board Statement:** Not applicable.

**Informed Consent Statement:** Not applicable.

**Data Availability Statement:** The data presented in this study are available on request from the corresponding author.

**Conflicts of Interest:** The authors declare no conflict of interest.

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
