# Peer review of "Low-Level Laser Irradiation Stimulates RANKL-Induced Osteoclastogenesis via the MAPK Pathway in RAW 264.7 Cells"

_applsci, doi:10.3390/app11125360_

Round 1

Reviewer 1 Report

The manuscript entitled ‘Low-level laser irradiation stimulates RANKL-induced osteoclastogenesis via the MAPK pathway in RAW 264.7 cells’ describes the effects of Ga-As-Al laser on murine cells, and its potential application to enhance osteoclastogenesis in bone diseases where OC activity might be suppressed.

The manuscript is well written, with clear statements and a careful state-of-art detailed introduction. Relevant literature is cited within the introduction as well as discussed and cross-checked meticulously in the discussion section, comparing the results obtained to previous works. The methodology might need some minor amendments to help understanding, as some verbs are missing and some rephrasing is needed. However, the methods section is well described and ordered to follow the manuscript. The results presented are systematic and described and discussed precisely. I would recommend the manuscript publication after some minor corrections.

I commend the authors for the work, the clear presentation of the results and the discussion, giving a clear overview of the field, and the current works and challenges.

General comments:

The authors might want to homogenize the units with separation or not from the preceding numbers. They are mixed in the manuscript.

Reference 1 contains the doi link which does not appear in the rest. Authors might want to double check that the citation style is correct.

Materials and methods, table 1: It seems no differences are between group 1 and 2, could it be that group 2 contains 10ng/ml of RANKL?

Section 2.4., line113: The authors might want to double check that cell membrane permeabilization was performed only for 1min and not 10min as described for immunostaining section.

A conclusion section with a paragraph highlighting the main outcomes from the study and the relevance in the field may help to strengthen the manuscript and improve the impact in the reader.

Some minor grammar details:

Abstract, line 14: The authors might want to expand the LLLT acronym for its first appearance.

Introduction, line 40: As a results, it leads…

Section 2.2, line 94-95: LLLT irradiation was at a wavelength…following a previously described protocol

Section 2.3, line 100: Use of after together with then sounds a bit redundant.

with of- please rewrite

line 104: this sentence is incorrect, please rephrase (proliferation rates of completely dissolved purple formazan crystals was measured…

section 2.6, line 127:Please rephrase this sentence, some verb is missing or wrongly spelled.

line 139: was used

Section 2.7, line 146: RANKL was

Sectio 2.8, line 159: RNA was isolated from the RAW264.7 cells treated with LLLT and RANKL using….

Section 2.9, line178: …were repeated for at least…

Page 5, line 203: Please rephrase this sentence, check English grammar.

Section 3.2, line 219: Figure 3, I would suggest to remove the dot after Figure

Discussion, page 9, line 332: I would suggest to add cathepsin K gene ‘expression’

Page 10, line 388: bone diseases

Reviewer 2 Report

The paper from Song et al. shows the effect of low-level laser irradiation on RAW 264.7 cell osteoclastogenesis. The paper has weak points and it must be improved in order to be published.

The abstract should be changed as it is just a summary of results (just a part of it).

Line 49: Calcitonin receptor is not a proteolytic enzyme.

Line 127: there is a mistake in the word “harv”..maybe you wanted to write harvested

Figure 2C: the pits are difficult to visualize. Improve the quality of the figure.

Figure 3: the quality of the figure must be improved.

Figure 4: Please, show also a membrane staining (i.e., actin staining) to show that the cells that you are showing are mature multinucleated osteoclasts.

Figure 5 and 6: In some panels (Fig. 5A and 6B) GAPDH misses. It should be showed. Moreover, in Fig. 5B ERK, JNK, p38 and GAPDH signals are saturated. Western Blots with no saturated signals should be presented. In Figure 6A GAPDH signals are saturated.

The discussion should be improved including comments about Figure 1, 2 and 3.

The English language and style should be carefully revised. 

Reviewer 3 Report

The authors described that LLLT irradiation to RAW264.7 cells treated with 10 ng of RANKL was effective to stimulate osteoclast differentiation. LLLT has been shown to stimulate biological activity of many kinds of cells and application of LLLT to the bone is curious. Many reports which applied LLLT to stimulation of bone formation have been published. However, the effect of LLLT irradiation to osteoclastogenesis is limited, and results of this study is interesting.

Why the authors stimulate RAW264.7 cells with 10 ng/mL of RANKL. This is the lowest dose of induction of osteoclasts. I think this dose is too low to evaluate the effect of LLLT on osteoclastogenesis. Fig. 3 showed that the expression of NFAT-c1 was upregulated about 1.2 times in 10 mg/mL RANKL-treated RAW cells compared with untreated RAW cells. It did not show clear response to osteoclastic differentiation. I think the authors should show data of LLLT irradiation to RAW cells untreated and 50 or 100 ng/mL of RANKL-treated cells in addition to RAW cells treated with 10 ng/mL of RANKL.

The authors should show the number of osteoclasts in Fig. 2B rather than % ratio.

The representative data of confocal microscopy looks interesting. However, it is not clear that nuclei of RANKL (+) and RANKL (+)/LLLT (15s) derive from multinucleated osteoclasts. They should be nuclei of multinucleated osteoclasts.

Round 2

Reviewer 2 Report

The authors clarified every doubt. The paper is ready for publication. 

Reviewer 3 Report

Results of this study were not always notable and the efficacy of stimulation of osteoclastogenesis was not so high. However, it may provides one of the new biological functions of LLLT.